# Geospatial Web Services Discovery through Semantic Annotation of WPS

**Meriem Sabrine Halilali** [1,*] , **Eric Gouardères** [2] , **Mauro Gaio** [1] **and Florent Devin** [3]

1    LMAP, Pau University (UPPA), 64013 Pau, France; mauro.gaio@univ-pau.fr
2    IPRA, Pau University (UPPA), 64013 Pau, France; eric.gouarderes@univ-pau.fr
3    Dataiku, 64000 Pau, France; florent.devin@dataiku.com
*    Correspondence: meriem-sabrine.halilali@univ-pau.fr

**Abstract:** This paper presents an approach to GWS (Geospatial Web Service) discovery through the semantic annotation of WPS (Web Processing Service) service descriptions. The rationale behind this work is that search engines that use appropriate semantic-based similarity measures in the matching process are more accurate in terms of precision and recall than those based on syntactic matching alone. The lack of semantics in the description of services using a standard such as WPS prevents the use of such a matching process and is considered a limitation of GWS discovery. The GWS discovery approach presented is based on the consideration of semantics in the service description method and in the matching process. The description of services is based on a semantic lightweight meta-model instantiated in the WPS 2.0 standard, extending the description of the service through metadata tags. The matching process is performed in three steps (functionality matching step, I/O (Input/Output) matching step and non-functional matching step). Its core is a semantic similarity measure that combines logical and non-logical matching methods. Finally, the paper presents the results of an experiment applying the proposed discovery approach on a GWS corpus, showing promising results and the added value of the three-step matching process.

**Keywords:** geospatial web service; semantic annotation; semantic web service description; semantic web service discovery; semantic web service matching; non-functional matching; WPS

## 1. Introduction

Today, with the development of service-oriented science and cloud computing technology, shared geodata is gradually being replaced by shared geographic information services [1].

Such technological improvements have required an update of current standards. In particular, the replacement of ISO (International Organization for Standardization) 19119 : 2005 for geographic information services by a new version ISO 19119 : 2016 [2] and OGC (Open Geospatial Consortium)'s WPS 1.0 by the version WPS 2.0 [3]. ISO 19119 : 2016 now provides a taxonomy of GWS, categorizing their functionalities, and thus providing support for the publication and discovery of GWS. A WPS specification defines the interface of a web service that allows for a description of the syntactic aspects of the functional properties in an open and recognized format. Thus, it can be used to support syntactic interoperability.

Service discovery is the process of selecting one or more services that meet a user's needs, taking into account functional and non-functional properties. A search is performed in a directory or catalogue of services in response to a user request. Using similarity measures, matching operations are performed between the attributes of the request and the properties stored in the service descriptions. Effective discovery relies on search engines with refined indexing and scoring techniques.

These techniques cannot only rely on syntactic aspects: they also need semantic aspects of the functional properties. The lack of semantics in service description using the WPS

standard prevents the use of such a matching process and is therefore considered as one of the main limitations of GWS discovery [4–6].

The process of adding semantic information may be conducted using annotations linked to formally specified vocabularies. Vocabularies are defined in ontologies in order to capture the meaning of the content and allow for logic-based reasoning. Using a reasoning engine and an appropriate similarity measure, we can then accurately match semantic queries to semantically described services, ensuring a greater precision and recall of search results. In addition, the choice of an appropriate semantic similarity measure is of great importance to ensure a good performance in terms of precision and recall rates.

This paper firstly proposes an approach with a fully operational proposal improving the WPS 2.0 standard, with an annotation principle taking into account semantic aspects of functional properties and describing non-functional properties. Secondly, the approach improves geospatial web services discovery with a new semantic similarity measure and a relevant matching process.

The remainder of this paper is organized as follows. Section 2 gives an overview of methods for describing and discovering GWS. Next, Section 3 outlines the two parts of the proposed approach. Section 3.1 presents a meta-model and its instantiation method to improve the WPS 2.0 standard. Section 3.2 presents a hybrid matchmaking method and a three-step matching process. Section 4 describes and discusses the architecture and experimentation of the proposed approach by presenting experiments, including GWS, used in the context of the Choucas project (http://choucas.ign.fr/, (accessed on 20 June 2021). This interdisciplinary research project involves researchers in geographic information sciences, computer science and automatic language processing. The project's purpose is heterogeneous data integration and spatial reasoning for localizing victims in mountain area) [7] and sets of other diversified services. Finally, Section 5 concludes this paper.

## 2. Related Work

GWS standards tend to provide a number of specifications or other specific criteria designed to be used consistently as a rule, guideline or definition. OGC and ISO have jointly developed the ISO 19119 : 2016 standard [8]. This standard provides a taxonomy of GWS, classifying their functionalities, one of the goals of which being to facilitate their publication and discovery. Unfortunately, the proposed taxonomy, beyond remaining too generic, was not designed to be easily automatically processed. Simultaneously, the OGC WPS [3] is a standard that facilitates the description of geospatial data processing processes on the web, such as: geocoding, route calculation, elevation profile, etc. A WPS specification defines the interface of a web service that allows for the text-based syntactic description of the service functionality and type of parameters in an open and well-known machine-readable format.

As previously stated, ISO 19119 : 2016 and OGC WPS 2.0 (i.e., the most recent versions of these standards) do not yet include the possibility of providing semantic information in their service description.

### 2.1. Semantic Description of Services

The semantic description of services has its origins in the Semantic Web [9]. Its role is to give meaning to data and services, helping to understand and use them correctly. The use of metadata enables the encoding of semantics in data and service descriptions. Technologies such as the RDF (Resource Description Framework) and the OWL (Ontology Web Language) can be used to structure metadata containing the description of concepts, relationships between entities or the categorization of data or services. Based on such technologies, two main approaches are proposed in the literature to describe semantically functional properties of web services: the semantic annotation approach and semantic language approach. In the former approach, the most important standard is SAWSDL (Semantic Annotations for WSDL) [10]. It specifies extension points for W3C-compliant web service metadata encoded with the WSDL (Web Service Description Language). In

the latter approach, OWL-S (Ontology Web Language for Service) [11] and WSMO (Web Service Modeling Ontology) [12] are the dominant ones. OWL-S and WSMO present ontologies as a language for describing services. The two different approaches have been used consistently in several works [13–16] dealing with the semantic description of GWS. However, being exclusively interested in the purely semantic aspects of services, these works have all omitted the use of the WPS standard for the syntactic aspects.

Thus, a challenge that has not yet been met is to merge semantic web techniques (ontology, semantic annotation, etc.) with the current geospatial standards. In [17], authors meet a part of this challenge: they introduce the idea of a *transparent semantic enablement layer* and its integration with OGC services. Despite its conceptual clarity, this approach did not find concrete expression due to the lack of implementation and evaluation of the proposed solution.

### 2.2. Functional Matching

In order to ensure GWS discovery, semantic-based frameworks have been proposed [18–20] to address functional matching. The most important step in the process of functional matching is to calculate the similarity of the I/O parameters between the services and the user request to find matched ones. In most cases, the I/O parameters of all services in the catalogue are exhaustively compared with the I/O parameters of the user request. I/O parameter matching can then be performed for services that are irrelevant from a functionality point of view, which is time-consuming. To avoid exploring irrelevant services, and thus decrease the response time, Refs. [19–21] added service functionality as a first criterion in the matching process. Therefore, they defined a domain vocabulary to classify geospatial web services functionalities. Services are labelled using this vocabulary, allowing us to perform a first matching step to quickly select relevant services. Then, the I/O matching step can be applied on selected services to refine the result. In [19,20], despite the implementation of semantic matching methods, the used semantic similarity measures remain naive, and only the proposal in [21] uses mechanisms specific to semantic matchmaking approaches. Existing work on semantic matchmaking can be grouped into three main categories: logic-based approaches, non-logic-based approaches and hybrid approaches. Logic-based approaches [22,23] use ontology concepts and logical rules to check the compatibility between the request and the service. They are mainly based on semantic matching filters called DoM (Document Object Model) filters. Non-logic-based approaches [24] aim to reduce the complexity of matchmaking by analyzing service descriptions based on information retrieval techniques, such as natural language processing, data mining, graph matching or computational mechanisms, for the numerical distance between concepts on given ontologies, such as the Wu and Palmer similarity measure [25]. Hybrid matchmakers [26–28] combine the advantages of non-logic-based techniques with the reasoning capabilities of logic-based techniques. In [21], the authors have adopted a hybrid approach that combines a deductive matching with a definition of specific logical filters and, on the other hand, a matching based on similarity mainly based on relations using the linguistic principles of synonymy, hyperonymy, hyponymy, etc. This is a very appropriate approach to resolve the mismatch between the terminology of a non-specialist user and that used by specialists to label services. Therefore, there may be concepts that are semantically close without being linked by any of the above relationships, and, therefore, they may not take into consideration services that may meet the user's need. Nevertheless, the use of this category of relationships remains very relevant, especially if used in a complementary way with other categories of relationships.

### 2.3. Non-Functional Matching

In the context of GWS discovery, in order to refine functional matching results, a number of geospatial research studies [29–32] take into account the non-functional matching step, often referred to as constraints satisfaction, preferences satisfaction or, also, the service recommendation step. The non-functional matching step exploits non-functional properties

(availability, cost, database quality of gazetteer, etc.), which are most often QoS (Quality of Service) attributes. This step is used to assist the user in their selection. It follows the functional matching step if it proposes several services that are functionally similar. The non-functional matching methods are varied. They are closely related to the non-functional attributes considered and the evaluation model used (real numbers, interval numbers, linguistic expression, ontology concepts, etc.). To our knowledge, a critical requirement today is to consider the non-functional properties of geographic information according to the different categories of geospatial services.

A first challenge is, therefore, to allow the current version of WPS to integrate semantic aspects in the description of the services' functionality, thanks to the technologies designed for the Semantic Web.

A second challenge is to improve the service discovery process by considering advances in semantic matchmaking to ensure a greater precision and recall of search results.

Finally, in addition to the functional properties, it is relevant to take into account the non-functional properties in order to recommend to the user a more refined ordered list of the most appropriate services.

## 3. Proposed Methods

In order to improve the GWS description, we propose a meta-model drawn from an abstraction of the semantic description of GWS. The improvement of discovery processes is based on a three-step matching that combines a semantic functionality match, semantic I/O match and non-functional match.

The meta-model was designed by adopting the same annotation philosophy as SAWSDL. However, our proposal is not limited to the coarse-grained annotation of web service descriptions with domain ontology concepts without specifying the role of these concepts in web service descriptions. For example, which concept signifies (annotates) the service category, which concept signifies (annotates) the I/O and which concept signifies (annotates) the QoS.

WPS 2.0 defines a *process* model to provide a syntactically interoperable description of geospatial processing functions. According to [3]:

'A *process* is a function that for each input returns a corresponding output; a *process description* is an information model that specifies the interface of a process and it is used to register the service in a catalog as a model for geospatial service discovery; and an *abstract process model* specifies generic requirements for the description of the process. In other words it represents an abstraction of the description of the process'.

Thus, the focus for improving the geospatial service discovery mechanism is on improving the *abstract process model* and the *process description*.

In accordance with this statement, the terms *service* and *process* will be used in an equivalent way to refer to a web service that provides a simple or complex geospatial processing operation that may contain multiple input and output parameters.

### 3.1. SAWPS (Semantic Annotation for WPS)

The method of describing GWS follows two steps: Section 3.1.1 designs a lightweight meta-model supporting the semantic description of functional properties and description of non-functional properties; Section 3.1.2 explains the protocol to instantiate our meta-model in WPS 2.0.

### 3.1.1. A Lightweight Meta-Model for Description of GWS

Figure 1 represents the main elements of the meta-model as an UML (Unified Modeling Language) class diagram.

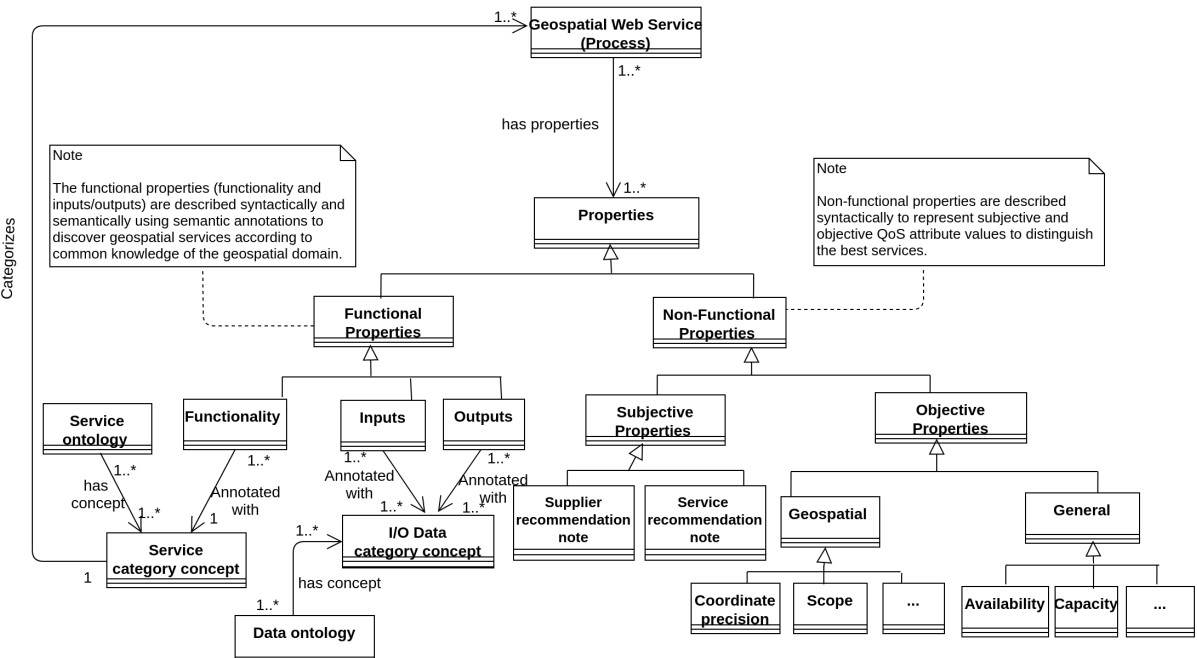

**Figure 1.** UML class diagram representing the meta-model of geospatial web service description.

As shown in the diagram, each service description contains information about both functional and non-functional properties. Functional properties mainly concern service functionality and information about its I/O parameters. The service functionality will be semantically annotated with a service ontology concept. This allows us, on one hand, to link the different services to common knowledge and, on the other hand, to automate and lighten the discovery process. To enable the application of semantic I/O matching, I/O has to be semantically annotated with the I/O data category concept according to a data ontology. Within the context of this work, we exclusively consider the relations of subsumption between the concepts of each ontology.

Finally, regarding the non-functional properties information represented in the meta-model, they are divided into two categories:

**Subjective properties**: represent the QoS according to user satisfaction levels. At this time, we are considering two recommendation notes. The first is a *service recommendation note*, which expresses users' reviews about the service. The second is a *supplier recommendation note*, which will be automatically calculated from the service recommendation notes of all services provided by that supplier;

**Objective properties**: represent the QoS according to *general properties* and *geospatial properties*. The first represents the QoS according to compliance with general requirements in terms of availability, reliability, etc., whereas the second represents the QoS according to compliance with the geospatial requirements. They differ according to the category of service under consideration. Some service categories require specific non-functional properties to be considered. For example, for visualization services, the zoom quality is of importance, whereas geocoding services depend on the quality of the gazetteers used for geocoding. The work proposed in [33] presented non-functional properties of some gazetteers and calculated the values of these properties. Table 1 presents examples of properties inspired by this work.

**Table 1.** Objectives non-functional properties, specific of geocoding services.

| Property | Property Description |
|----------|---------------------|
| Scope | Depends on the database used: whether it includes a communal, regional/national or global field. |
| Update frequency | Depends on how often the sources are updated: no update, annual, monthly, weekly or daily. |
| Coordinate precision | Depends on the coordinate system used: the precision can be metric, centimetric or decimetric. |
| Richness of annotation | Depends on the detail of the descriptive information. Annotations can be detailed in a major, moderate or minor way. |
| Lineage | Depends on the diversity of data sources. There can be one, two or more sources. |

### 3.1.2. WPS Instantiation

The current *abstract process model* of WPS presents only a syntactic description of service and I/O parameters (see Figure 2). As part of the description of the service, an attempt has been made by the standard to propose a new *service concept* (*process concept*).

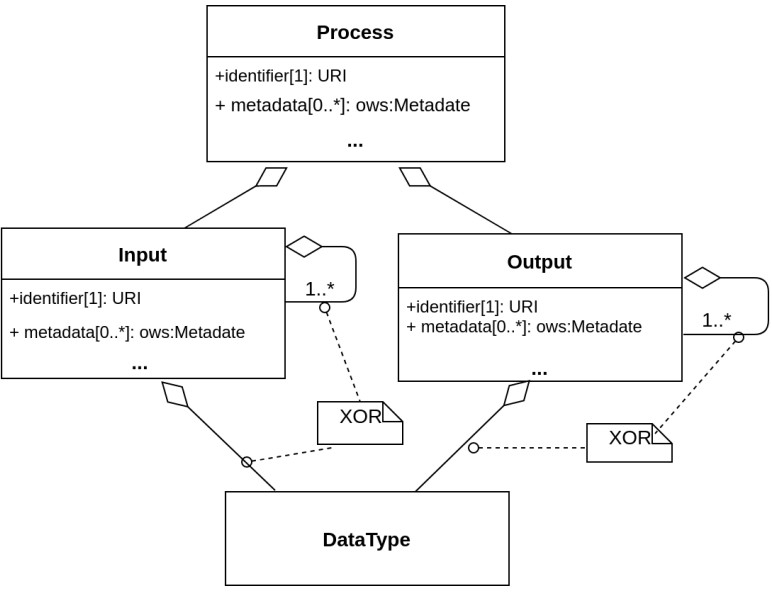

**Figure 2.** UML Class diagram describing the abstract process model [3].

A service concept (*process concept*) is an object added as service (*process*) metadata that provides documentation about a group of services (processes) (linked to an HTML (HyperText Markup Language) page or similar multimedia formats). However, it remains a syntactic description of service functionality compared to our proposal.

The proposed meta-model extends the *abstract process model* of WPS 2.0. In this way, as mentioned above, services can be described by functional properties (functionality and I/O) and non-functional properties (general and geospatial). An approach through annotation was chosen to instantiate the meta-model in the WPS description. Its implementation, based mainly on the use of the official *metadata* tag, ensures compliance with the standard. This tag has two attributes: *role* and *href* (*role* identifier indicates the role of the metadata and *href* references the metadata with HTTP (Hypertext Transfer Protocol)-URI (Uniform Resource Identifier) value type). Therefore, the annotations will be recorded in the metadata tag in the form of simple links with an appropriate role identifier. Table 2 (the presented HTTP-URI are not accessible on the web. They are mentioned as examples for illustration purposes) describes the defined new roles and the reused one (first line in the table).

**Table 2.** Extended roles for metadata tag for SAWPS (Semantic Annotation for WPS).

| Role | Definition | Example |
|---|---|---|
| http://www.opengis.net/ spec/wps/2.0/def/process/ description/documentation | Reference any documentation. This is an already existing role in WPS. | <ows:Metadata xlink:role="http:// www.opengis.net/spec/wps/2.0/def/ process/description/documentation" xlink:href="http://erig.univ-pau.fr/ PERDIDO/api.jsp"/> |
| http://choucas.ign.fr/ /spec/wps/2.0/def/ process-profile/concept_ functionality | Reference a service *process* concept linked to one of the concepts in a service ontology. Provide high-level reference about a generic group of services processes that have the same functionality. | <ows:Metadata xlink:role="http: //choucas.ign.fr/spec/wps/2.0/def/ process-profile/concept_functionality" xlink:href="http: //choucas/ontologie_service/2018/6 /untitled-ontology-6#Direct_ geocoding_service"/> |
| http://choucas.ign.fr/ spec/wps/2.0/def/process- profile/concept_Input | Reference an input data type concept related to one of the concepts of data ontology. | <ows:Metadata xlink:role="http: //choucas.ign.fr/spec/wps/2.0/def/ process-profile/concept_Input" xlink:href="http://choucas/ ontologie_data/2020/0/untitled- ontology-25#SpatialEntity"/> |
| http://choucas.ign.fr/ spec/wps/2.0/def/process- profile/concept_Output | Reference an output data type concept related to one of the concepts of data ontology. | <ows:Metadata xlink:role="http: //choucas.ign.fr/spec/wps/2.0/def/ process-profile/concept_Output" xlink:href="http: //choucas/ontologie_data/2020/0 /untitled-ontology-25 #GeographicCoordinates"/> |
| http://choucas.ign.fr/ spec/wps/2.0/def/process- profile/Non-functional_ properties | Reference a path that contains information about the non-functional properties related to the service. | <ows:Metadata xlink:role="http://choucas.ign.fr/ spec/wps/2.0/def/process-profile/ Non-functional_properties" xlink:href="http://choucas.ign.fr/ NonFunctionalDescription/ NFGetToponym.xml"/> |

For functional properties, a link to the appropriate concept of ontology is instantiated, whereas, for non-functional properties, the link will point to an XML (eXtensible Markup Language) instance containing the information (last line in the table).

### 3.2. GWSD (Geospatial Web Services Discovery)

The description of the GWS discovery method follows two steps. First, Section 3.2.1 defines a hybrid matchmaking method for measuring the semantic similarity of GWS. Then, Section 3.2.2 defines a three-step matching approach for GWSD (Geospatial Web Services Discovery). This approach combines a semantic service functionality match, semantic I/O parameters match and non-functional match.

### 3.2.1. A Hybrid Matchmaking Method

The proposed method, named *SimCalc*, calculates a similarity between the service consumer's requirements and the descriptions of service instances with regard to functionality and I/O parameters by combining a non-logical method and a logical method.

For describing geospatial web services, we use domain ontologies. These structures provide links that represent semantic information derived from the path lengths of knowledge networks.

The **non-logical method** common similarity measure of Wu and Palmer (*SimWP*) [25] has been selected because of its adoption in some recent research [34,35] and its efficiency and simplicity of implementation, while remaining as expressive.

In context, this measure can be used to calculate the semantic similarity between two concepts in an ontology based on the hierarchical structure of the ontology. The method is defined as follows:

Given an ontology $\Omega$ formed by a set of concepts and a root concept $R$, $C_1$ and $C_2$ represent two of its concepts, on which, the similarity will be calculated. The principle of the computation is based on the distances $N_0$, $N_1$ and $N_2$, separating root concept $R$, concept $C_1$ and concept $C_2$ from the closest common ancestor $CS$ (see Figure 3). The *SimWP* that assigns a score $\in [0, 1]$ is defined in Equation (1).

$$SimWP = \frac{2.N_0}{N_1 + N_2 + 2.N_0} \tag{1}$$

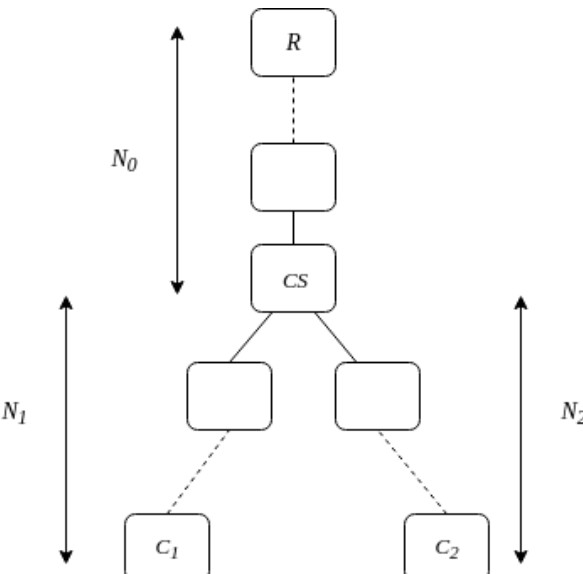

**Figure 3.** Wu and Palmer ontology example.

The measure obtained with this method depends only on the concepts' depth. Due to the fact that most ontologies have limited depths compared to the number of concepts, it can be seen that the method is processed in an acceptable time frame. However, this measure cannot be used directly to match web services since it is symmetrical (i.e., $SimWP(C1, C2) = SimWP(C2, C1)$); the types of concepts to match must be differentiated according to the role of the concept in the service description. Moreover, for comparing service parameters, the single use of such a method may cause a bias by its tendency to give priority to neighboring concepts (concepts having the same parent), rather than concepts belonging to the same hierarchy (a concept class and its sub-classes).

Therefore, to avoid this effect, we propose combining this method with a logical matching method.

**Logical methods** (i.e., logical-based matchmaking) have been used by several research studies to check whether the I/O parameters of a service are compatible with the I/O parameters of a request [22]. A common approach to logical-based matchmaking is to define a set of rules (filters) that dictate what kind of logical relationships is acceptable between the I/O parameters of a service and the I/O parameters of a request [27].

This kind of matching takes into account the entire I/O signature, so the degree of correspondence between a service and a request cannot be calculated. According to [26], a more flexible approach is required to be able to assess the degree of a match between a service and a request.

Consequently, we propose a logical-based matchmaking method based on individual links between parameters of the service and request. The links will be used for functionality and I/O discovery purposes. Given a set of concepts in an ontology $\Omega$, the logical link $LogLink(SP, RP)$ between a service parameter concept $SP$ and a request parameter concept $RP$ can belong to one of the five filter categories detailed in Table 3.

**Table 3.** Filter categories.

| Cat. Name | Definition | Formal Rep. |
|---|---|---|
| *Exact* | if $RP$ and $SP$ are equivalent | $\Omega \models RP \equiv SP$ |
| *Plug$-$In* | if $RP$ is a sub-class of $SP$ | $\Omega \models RP \sqsubseteq SP$ |
| *Subsumes* | if $RP$ is a super-class of $SP$ | $\Omega \models RP \sqsupseteq SP$ |
| *Fail* | if both $RP$ and $SP$ are specified but no relationship can be determined in any of the above ways | $\Omega \models SP \cap RP \sqsupseteq \perp$ |
| *Unknown* | if $RP$ or $SP$ or both are not specified | None |

Finally, in order to calculate the matching score between a $SP$ parameter and a $RP$ parameter, we propose a hybrid method that combines the two previous ones (logical and non-logical). We define *SimCalc* (see Equation (2)) as a function whose result is a score $\in [0, 1]$ obtained after the evaluation of two other functions (*LogLink* and *SimWP*) and taking into account the type of the parameter to match (*input*$-$*input* (*in*$-$*in*), *output*$-$*output* (*out*$-$*out*) or *functionality*$-$*functionality* (*func*$-$*func*)).

$$SimCalc(LogLink(SP, RP)) = \begin{cases} 1.0; & \text{if } LogLink(SP, RP) = Exact \\ SimWP(SP, RP); & \text{if } (LogLink(SP, RP) = Plug - In) \\ & \text{and } (in - in) \\ SimWP(SP, RP); & \text{if } (LogLink(SP, RP) = Subsumes) \\ & \text{and } ((out - out) \text{ or } (func - func)) \\ 0.0; & \text{if } (Loglink(SP, RP) = Fail) \\ 0.0; & \text{if } (Loglink(SP, RP) = Unknown) \\ 0.0; & \text{if none of the above conditions are observed.} \end{cases} \quad (2)$$

where:

$\{SimCalc(LogLink(SP, RP)) \in \mathbb{R} \ \ 0 \leq SimCalc(LogLink(SP, RP)) \leq 1\}$ the matching score between $SP$ and $RP$

$\{LogLink(SP, RP) \in \{Exact, Plug - In, Subsumes, Fail, Unknown\}\}$ the logical link between $SP$ and $RP$

The score assigned to an (*out*$-$*out*) parameter type or a (*func*$-$*func*) type depends on the request. We consider service functionalities (and outputs) that are the same or more specific than those mentioned in the request. Therefore, the method accepts only *exact* and *subsumes* filters to match these parameters. Inversely, since the score assigned to the (*in*$-$*in*) parameter type depends on the service, we consider service inputs that are the same or more generic than the ones mentioned in the request. Therefore, the method accepts only *exact* and *plug*$-$*in* filters.

As mentioned earlier, a hybrid matching approach provides the benefits of both logical and non-logical matching. The hybrid approach presented in [21] combines non-logical similarity matching based on a particular category of semantic relations in ontologies and logical matching with a definition of specific logical filters (subsumes, plugin, etc.). Therefore, from a methodological point of view, we have adopted a similar approach. The fundamental difference between the two approaches lies in the ontological properties retained for the calculation of similarity. In the case of the above paper, the similarity matching is based on the semantic relationships that may exist between the terminology used to identify services and that used by users. For this purpose, the authors used some well

known linguistic principles, such as synonymy, hyperonymy, hyponymy, etc. The matching score is computed as an aggregation of numerical scores, between 0 and 1, depending on the evaluated relations and their relevance to the matching process. In the current version of our proposal, we have preferred to base the evaluation of the similarity on the semantic subsumption relations that exist in the ontology. The Wu and Palmer similarity measure that we use computes a numerical score that represents the semantic similarity between two concepts of an ontology as a function of the depth of the concepts, which gives it the ability to compute the semantic similarity of the concepts in a faithful way, i.e., respecting the hierarchical representation of the knowledge in the ontology. We can thus compute a similarity between two concepts, even if there is no obvious semantic relationship defined between them in the sense of [21]. For example, let us consider "Geosemantic analysis" as a part of the requester requirements in terms of the desired categories of service and two concepts representing two classes of SWG categories linked by a subsumption relation in the service ontology, which are: "Geosemantic_analysis_service" and "NERC_service" (NERC is for named entity recognition and classification) (see Figure 7). According to our approach, the similarity score will be 0.88. While, from a terminological point of view, it appears to be rather complex to obtain, for the NERC service, one of the semantic relations in the sense of [21], the score should be close to 0. In fact, the two approaches do not address the same problem and it would be very interesting in the long term to be able to make them cooperate. Since both works use the same methods to compute similarity on functionality and on I/O parameters, the same observation could be made when considering two concepts from the data ontology, e.g., the concepts "offset" and "DistanceCategory".

### 3.2.2. Three-Step Matching Process

The first step is called **semantic functionality matching**. When a request is submitted, the service instances matching the requested functionality concept are then discovered. Only services whose score is equal to or greater than a given threshold $\theta \in [0, 1]$ are retained.

The matching is based on the proposed *SimCalc* function with a $func-func$ type restriction. For example, if the *geocoding* concept is requested, all service instances annotated with a concept that has a matching score $\geq \theta$ with a *geocoding* concept are retained at this step. The value of $\theta$ can be chosen from a number of matching categories (e.g., strict for $\theta = 1.0$, medium for $\theta = 0.5$ and fuzzy for $\theta = 0.0$). The functionality-based matching aims to quickly exclude large amounts of unrelated services.

The second step, called **semantic I/O matching**, is then applied to refine the result. The service descriptions are browsed to determine if their I/O properties meet the I/O properties defined by the service requester. We propose a method that allows for optimized matching according to the maximum score of the *SimCalc* function and the type of parameter. As in the previous step, only services whose score is equal to or greater than a given threshold $\theta \in [0, 1]$ are retained. The threshold value can be chosen from a number of matching categories.

Given $\Omega$ an ontology, $S$ a service and $R$ a request, and $IS$, $OS$ and $IR$, $OR$ finite sets of linked concepts of input and output parameters for $S$ and $R$, respectively, the *FunScore* function is defined to calculate the I/O matching score (see Equation (3a)).

The score assigned to an $input-input$ matching is based on the number of service inputs instead of demand inputs. The priority here is to satisfy the notion that required inputs for the service have to be met. However, it remains acceptable if one of the inputs specified by the request is not used. Therefore, the maximum matching score for each service input is summed and then divided by the number of input parameters required by the service (see Equation (3b)).

Inversely, the score assigned to an $output-output$ matching depends on the number of outputs specified in the request. The priority here is to satisfy the notion that required outputs for the request have to be met. Even in cases where the service generates some additional outputs, this remains acceptable. Therefore, the maximum matching score for

each request output is summed and then divided by the number of output parameters required by the request (see Equation (3c)).

The third step, called **non-functional matching**, extended the process for matching GWS, integrating contextual information (i.e., non-functional properties). In order to achieve this goal, we define *NonFunScore* as a non-functional matching function. This function searches among the set of candidate service descriptions selected in the previous steps and uses the non-functional properties proposed in the service description meta-model (see Section 3.1.1). Equation (4) calculates the non-functional score value of candidate services. In this equation, we use weight $\alpha \in [0, 1]$ in order to increase or decrease the objective and subjective properties consideration.

$$FunScore(S, R) = \frac{IMatch(IS, IR) + OMatch(OS, OR)}{2} \tag{3a}$$

$$IMatch(IS, IR) = \frac{\sum_{i=1}^{n} \max_{j=1}^{m}(SimCalc(Loglink(ISi, IRj)))}{n} \tag{3b}$$

$$OMatch(OS, OR) = \frac{\sum_{k=1}^{p} \max_{l=1}^{q}(SimCalc(Loglink(OS_l, OR_k)))}{p} \tag{3c}$$

where:

$\{FunScore(S, R) \in \mathbb{R} \ \ 0 \leq FunScore(S, R) \leq 1\}$ the functional score for $S$ in regard to $R$

$\{IMatch(IS, IR) \in \mathbb{R} \ \ 0 \leq IMatch(IS, IR) \leq 1\}$ the matching score between $IS$ and $IR$ of $S$ and $R$

$\{OMatch(OS, OR) \in \mathbb{R} \ \ 0 \leq OMatch(OS, OR) \leq 1\}$ the matching score between $OS$ and $OR$ of $S$ and $R$

$$NonFunScore(S) = \alpha.\frac{\sum_{i=1}^{n} ObjP(i)}{n} + (1 - \alpha).\frac{\sum_{j=1}^{m} SubP(j)}{m} \tag{4}$$

where:

$\{NonFunScore(S) \in \mathbb{R} \ \ 0 \leq NonFunScore(S) \leq 1\}$ the non-functional score for service $S$

$\{ObjP(i) \in \mathbb{R} \ \ 0 \leq ObjP(i) \leq 1\}$ the value of the objective property $i$

$\{SubP(j) \in \mathbb{R} \ \ 0 \leq SubP(j) \leq 1\}$ the value of the subjective property $j$

$\{\alpha \in \mathbb{R} \ \ 0 \leq \alpha \leq 1\}$ a value representing the weight

Finally, after the three matching steps, final assessment values of services are determined. Here, we refer to the final assessment as the basic recommendation score (*RecScore*). The computation of the *RecScore* value uses both the functional score (result of *FunScore* function) and non-functional score (result of *NonFunScore* function) as described in (Equation (5)). In this equation, we use weight $\beta \in [0, 1]$ in order to increase or decrease the functional matching consideration. The weight allows us to give the user the possibility to parameterize the final score since the interest of the users concerning the non-functional aspect is different: some prefer to take it into account and others do not.

$$RecScore(S) = \beta.FunScore(S, R) + (1 - \beta).NonFunScore(S) \tag{5}$$

where:

$\{RecScore(S) \in \mathbb{R} \ \ 0 \leq RecScore(S) \leq 1\}$ is the recommendation score of service $S$

$\{\beta \in \mathbb{R} \ \ 0 \leq \beta \leq 1\}$ a value representing the weight

Nevertheless, taking into account the non-functional properties, even in a basic way, gives a real added value in the final proposal for the ranking (in order of preference) of the discovered services.

## 4. Architecture and Experimentation

### 4.1. Architecture

The contribution of this paper is twofold: (1) allowing for the creation of a semantic description for WPS according to the proposed annotation method (SAWPS); (2) supporting the discovery of a geospatial web service (GWSD) based on annotated descriptions. The functional architecture of the proposed SAWPS and GWSD approaches (illustrated in Figure 4) is composed of five modules :

1.  *WPS file annotation*: the entry points of this module are the domain ontologies, a WPS file syntactically describing the functional properties of a geospatial service and information about non-functional properties (QoS). This module applies the annotation approach described in Section 3.1. It subsequently allows for the creation of a directory of annotated WPS service descriptions;
2.  *SAWPS analyzer*: handles the annotation concept extraction of functionality and I/O associated with the semantically annotated WPS descriptions;
3.  *Semantic matching*: performs semantic matching between the user request and the set of WPS services by comparing different concepts according to the proposed method (see Section 3). It uses the domain ontologies and the Jena API reasoner (Jena: https://jena.apache.org/, accessed on 12 January 2020) to infer relationships between concepts;
4.  *QoS file analyzer*: handles the extraction of non-functional property (QoS) values from the description files;
5.  *QoS calculation*: calculates a non-functional score (QoS score) for each service discovered by the semantic matching module.

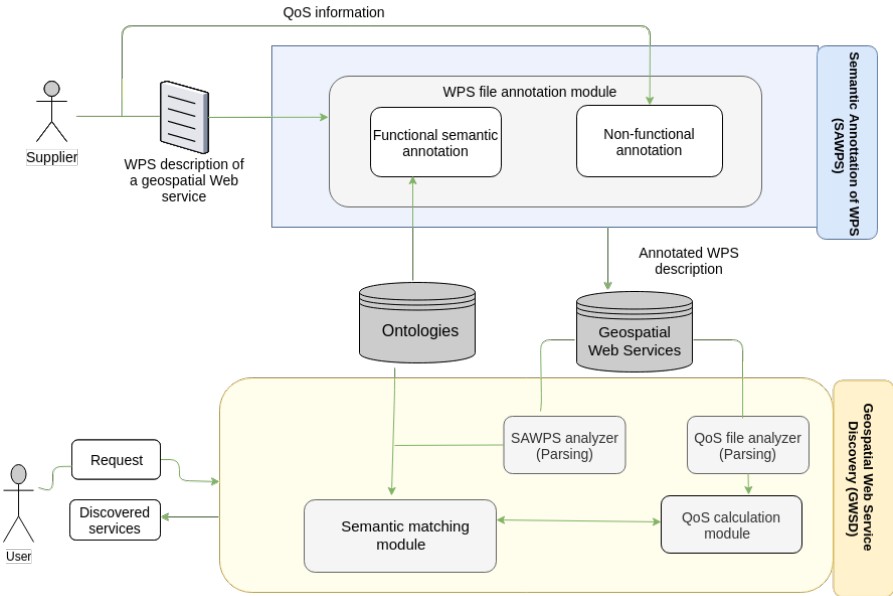

**Figure 4.** SAWPS and GWSD architecture.

### 4.2. SAWPS Services Corpus

To the best of our knowledge, there is neither a semantically described corpus of GWS nor a platform available to test and evaluate GWS semantic discovery methods. Therefore, to experiment and to discuss our contributions, we generated a corpus of 94 services described as WPS processes according to the WPS 2.0 description model. Then, we semantically described these processes following the SAWPS annotation approach.

To obtain a diversified corpus, we selected a set of services with different characteristics: functionalities, types of I/O parameters, number of I/O parameters, etc. All of the selected services are related to the geospatial domain. First, we selected 4 services that have been developed within PERDIDO project (PERDIDO Geoservices: http://erig.univ-pau.fr/PERDIDO/api.jsp, accessed on 12 June 2021) and used in the context of the Choucas project for itinerary reconstruction from a text, including services for POS tagging, for identifying spatial entities, for geocoding and finally a service for geographic format conversion. Then, we selected 2 services proposed by the IGN (National Institute of Geographic and Forest Information) (IGN Geoservices: https://geoservices.ign.fr/documentation/geoservices/, accessed on 12 June 2021) offering functionalities for autocompletion and itinerary calculation between two points. We also selected 4 services that use 4 different gazetteers cited in [33]. We analyzed the functionalities and I/O of these services and then we described variants of these services according to the WPS 2.0 syntactic description model (We used a program of the 52° North/javaPS to generate the syntactic description files automatically). To enrich our corpus, we retrieved 84 open source WPS *process* descriptions published by the 52° North project [36] (52° North WPS Processes: https://github.com/52North/WPS/tree/dev/52n-wps-webapp/src/main/webapp/examples/localWPSFiles/xmlDescriptions, accessed on 12 June 2020). They propose several functionalities related to the geospatial domain, such as distance calculation, surface interpolation, etc. We updated these descriptions to the current version 2.0 of WPS. Finally, the process of semantic annotation was carried out. In this context, the approach proposed in Section 3.1 allows for flexibility in the choice of the ontologies used to annotate the functionalities and I/O of services. To perform our experimentation, we designed two domain ontologies, service ontology and data ontology, in order to semantically annotate the service functionality and the I/O of a service. For the two designed ontologies, we exclusively considered the subsumption relations between the concepts. Another principle adopted when designing service ontology was to ensure a good balance between a generic categorization to promote reuse and a specific categorization to address different business needs, including the business needs within the Choucas project. Therefore, the method followed to design this ontology can be summarized in two main steps: the first one was the formalization of the core vocabulary (generic categories of geospatial services) based on the categories proposed by ISO 19119 : 2016 revision. Secondly, we enriched the vocabulary with specific service categories and linked them to the generic categories. Concerning data ontology, the Choucas project is in partnership with IGN, the French public state administrative establishment that produces and maintains geographical information for France. In this context, during the design of the ontology core, we reused a maximum of concepts proposed by our partner in order to promote reuse. Then, the ontology was enriched with other concepts related to the different services used in our experimentation.

Finally, we also referred to and drew inspiration from a set of works in the literature that define a vocabulary that can be used to semantically describe the I/O of these services. In particular, we are interested in the work of the Choucas project, which defines and specialises a vocabulary based on the TEI description language [37], as well as other work of [38] that provides an ontological formalism of the geography markup language (GML) format. In addition, for each service, an annotation was realized for linking the service to an XML file that represents the non-functional properties. The values corresponding to the non-functional properties were simulated, since this information is not yet available, except for geocoding services, which use gazetteers. For these services, a set of real values of objective properties specific to geocoding services was considered (see Table 1). The values were extracted from the work presented by [33].

Figure 5 illustrates an example of a semantic description of a geocoding service.

```
<wps:Process>
    <ows:Title>Geocoding</ows:Title>
    <ows:Abstract>This service returns the list of toponyms found in the input text with their geolocation. The input text must be
annotated with spatial entities</ows:Abstract>
        <ows:Identifier>http://choucas.ign.fr/profileregistry/implementation/GetToponym</ows:Identifier>
        <ows:Metadata xlink:role="http://choucas.ign.fr/spec/wps /2.0/def/process-profile/concept_functionality" xlink:href="http://
choucas/ontology service/2018/6/untitled-ontology-6#Direct geocoding service" />
        <ows:Metadata xlink:role="http://www.opengis.net/spec/wps/2.0/ def/process/description/documentation" xlink:href="http://
erig.univ-pau.fr/PERDIDO/api.jsp" />              Semantic annotation of service functionality
    <wps:Input minOccurs="1">
        <ows:Title>Content spatial entities</ows:Title>
        <ows:Metadata xlink:role="http://choucas.ign.fr/spec/wps /2.0/def/process-profile/concept_Input" xlink:href="http://choucas/
ontologie data/2020/0/untitled-ontology-25#SpatialEntity" />
        <ows:Abstract>Text with spatial entities</ows:Abstract>      Semantic annotation of the service input
        <ows:Identifier>content</ows:Identifier>
        <wps:ComplexData>
            <wps:Format mimeType="text/xml" encoding="UTF-8" schema="http://choucas.ign.fr/Schema-xml/Schema-NER.xsd" />
        </wps:ComplexData>
    </wps:Input>
    <wps:Output>
        <ows:Title>content coordinates</ows:Title>
        <ows:Metadata xlink:role="http://choucas.ign.fr/spec/wps /2.0/def/process-profile/concept_Output" xlink:href="http://
choucas/ontology data/2020/0/untitled-ontology-25#GeographicCoordinates" />
        <ows:Abstract>List of toponyms with geolocation (geographic coordinates)</ows:Abstract>
        <ows:Identifier>content</ows:Identifier>      Semantic annotation of the service output
        <wps:ComplexData>
            <wps:Format mimeType="text/xml" encoding="UTF-8" schema="http://choucas.ign.fr/Schema-xml/Schema-GetToponym.xsd"
default="true" />
        </wps:ComplexData>
    </wps:Output>
    <ows:Metadata xlink:role="http://choucas.ign.fr/spec/wps /2.0/def/process-profile/Non-functional_properties"
xlink:href="http://choucas.ign.fr/NonFunctionalDescription/NFGetToponym.xml" />
</wps:Process>      Annotation to reference non-functional properties
```

**Figure 5.** How SAWPS allows us to encode the semantic annotation.

*4.3. GWSD Experimentation and Observations*

4.3.1. Experimentation

In order to discuss the proposed discovery approach, two different configurations were tested:

1.  A configuration with two matching steps, functionality matching and I/O matching;
2.  A configuration with three steps, functionality matching, I/O matching and non-fonctionnal matching (QoS score calculation).

In order to evaluate these two configurations, we applied, on the corpus of 94 services, a set of tests consisting of 10 semantic requests for 8 different service functionalities. We also vary acceptance thresholds (see Section 3.2.2):

- For functionality matching, a strict threshold $\theta = 1.0$, an average threshold $\theta = 0.5$ and a fuzzy threshold $\theta = 0.0$;
- For I/O matching, a strict threshold $\theta = 1.0$, an average threshold $\theta = 0.5$ and a threshold between the two $\theta = 0.7$.

When presenting the results for the functionality matching, the results for the 0.0 threshold allowed us to simulate the disregard of this step before the I/O matching step, whereas, for I/O matching, the results for thresholds <0.5 are not shown because, below 0.5, the study is not relevant, as there is a risk of ending up with a large number of false positives.

In addition, we used the precision and recall metrics and the average response time of a given requests. Finally, the second configuration requires the calculation of the non-functional score (QoS).

In the following, Table 4 contains 2 of 10 semantic requests from our set of tests and Tables 5 and 6 their associated results.

**Table 4.** Requests *R1* and *R2*.

| Name | Functionality | Inputs | Outputs |
|------|---------------|--------|---------|
| *R1* | Distance calculation service | SourceLocation, Impedance | Distance, EuclidianDistance |
| *R2* | Direct geocoding service | SpatialEntity | GeographicCoordinates |

**Table 5.** Results of request *R1*.

| Service | Functionality | Score | Inputs | Outputs | Score |
|---------|--------------|-------|--------|---------|-------|
| Euclidean Distance | Distance calculation service | 1.0 | SourceLocation | EuclideanDirection, EuclideanDistance | 0.97 |
| Euclidean Direction | Distance direction calculation service | 0.90 | SourceLocation | EuclideanDirection, EuclideanDistance | 0.97 |
| Euclidean Allocation | Distance zone calculation service | 0.90 | SourceLocation | EuclideanDirection, EuclideanDistance, EuclideanAllocation | 0.97 |
| Cost Distance | Distance calculation service | 1.0 | SourceLocation, Impedance | CostDistance, CostBackLink | 0.72 |
| Cost BackLink | Distance neighbor calculation service | 0.90 | SourceLocation, Impedance | CostDistance, CostBackLink | 0.72 |
| Cost Allocation | Distance zone calculation service | 0.90 | SourceLocation, Impedance | CostAllocation, CostDistance, CostBackLink | 0.72 |
| Corridor | Distance sum calculation service | 0.90 | CostDistance, CostDistance | Corridor | 0.22 |
| Cost Path | Distance path calculation service | 0.90 | DestinationLocation, CostDistance, CostBackLink | CostPath | 0.0 |

The first request (*R1*) corresponds to the search for a `distance calculation service` and the second request (*R2*) corresponds to the search for a `direct geocoding service`. Remember that only the geocoding service has real values for non-functional properties, which is why the *R2* query was chosen here. The *R1* query represents the situation where the values for the non-functional properties have been simulated. In the corresponding tables, the three columns show the functionality and the I/O of the required service. These tables are accompanied by extracts from the service ontology (see Figures 6 and 7) and extracts from the data ontology (see Figures 8 and 9) in order to show, for the presented requests, the concepts related to the functionalities and I/O. The figures show the class hierarchy in the ontologies described with `is-a` relations between classes and sub-classes. The orange color shows the equivalence of two classes. The purple and green color on the relationship arrows show the child and parent classes, respectively, of a selected class. The selected class appears surrounded by a blue border.

**Table 6.** Results of request *R2*.

| Service | Functionality | Score | Inputs | Outputs | Score |
|---------|--------------|-------|--------|---------|-------|
| GeoNames | Direct geocoding service | 1.0 | PlaceName [a] | GeographicCoordinates | 1.0 |
| OS50k | Direct geocoding service | 1.0 | PlaceName | GeographicCoordinates | 1.0 |
| SwissNames | Direct geocoding service | 1.0 | PlaceName | GeographicCoordinates | 1.0 |
| TGN | Direct geocoding service | 1.0 | PlaceName | GeographicCoordinates | 1.0 |
| GetToponyms | Direct geocoding service | 1.0 | SpatialEntity | GeographicCoordinates | 1.0 |
| ConvPerdido | Geographic format conversion service | 0.0 | GeographicCoordinates | GeographicCoordinates | 0.5 |
| Autocompletion | Autocompletion Service | 0.0 | SpatialEntity | Adress | 0.5 |

[a] PlaceName and SpatialEntity are equivalent concepts (see Figure 9).

After that, we present the associated results (see Tables 5 and 6) with the names of the services, their functionalities, the matching score between the required and the offered functionality, the I/O and the matching scores between the required and the offered I/O.

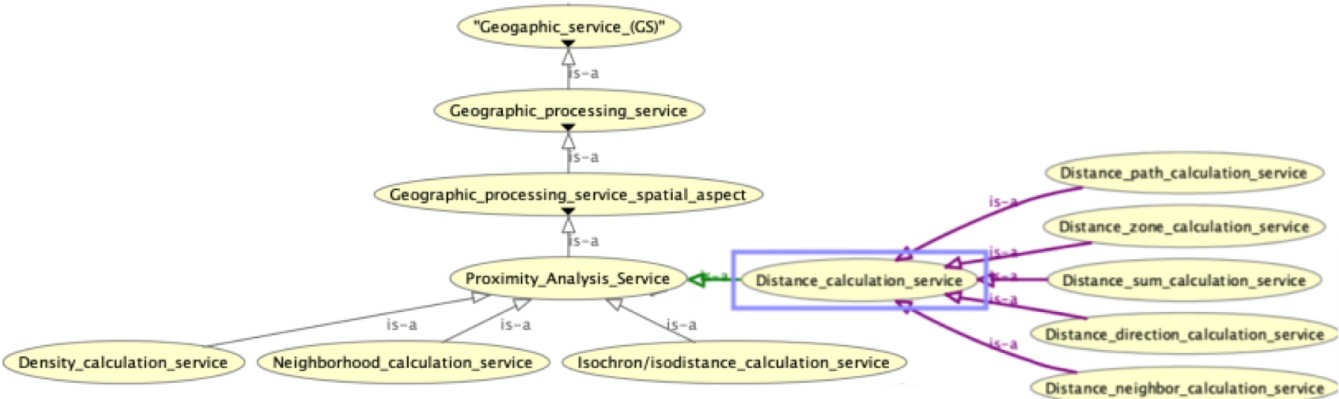

**Figure 6.** Extract from the service ontology for the *R1* request example.

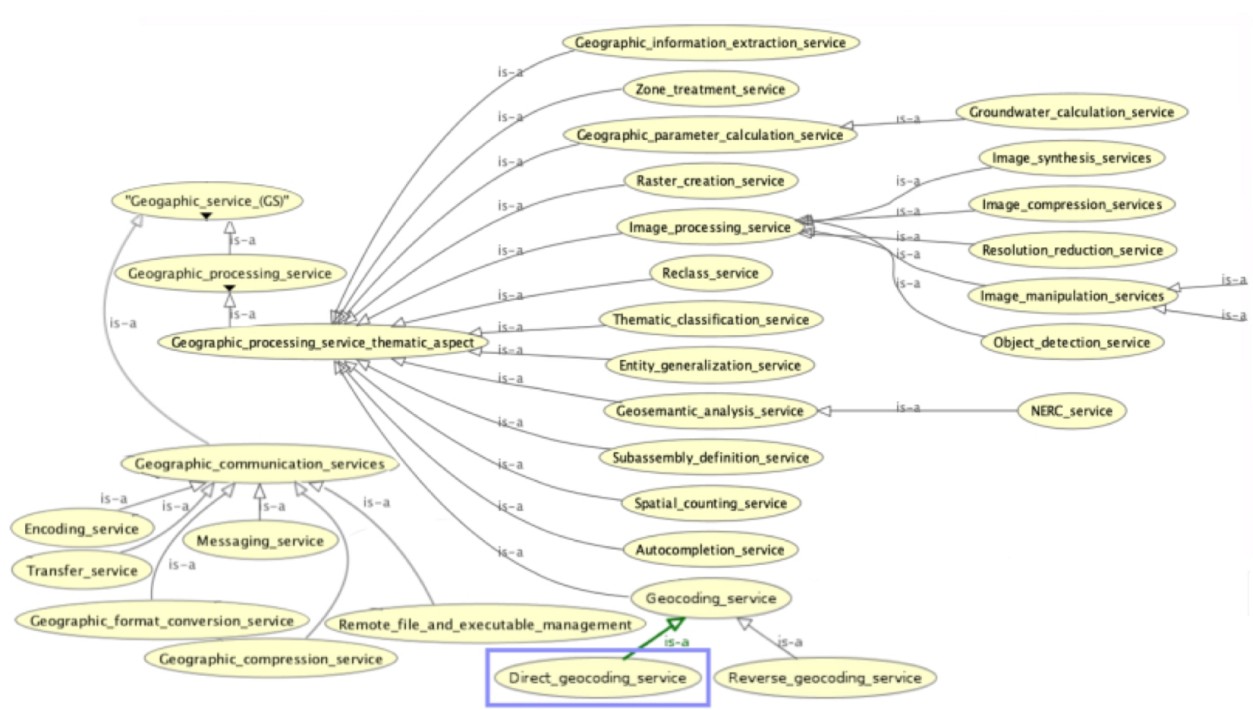

**Figure 7.** Extract from the service ontology for the *R2* request example.

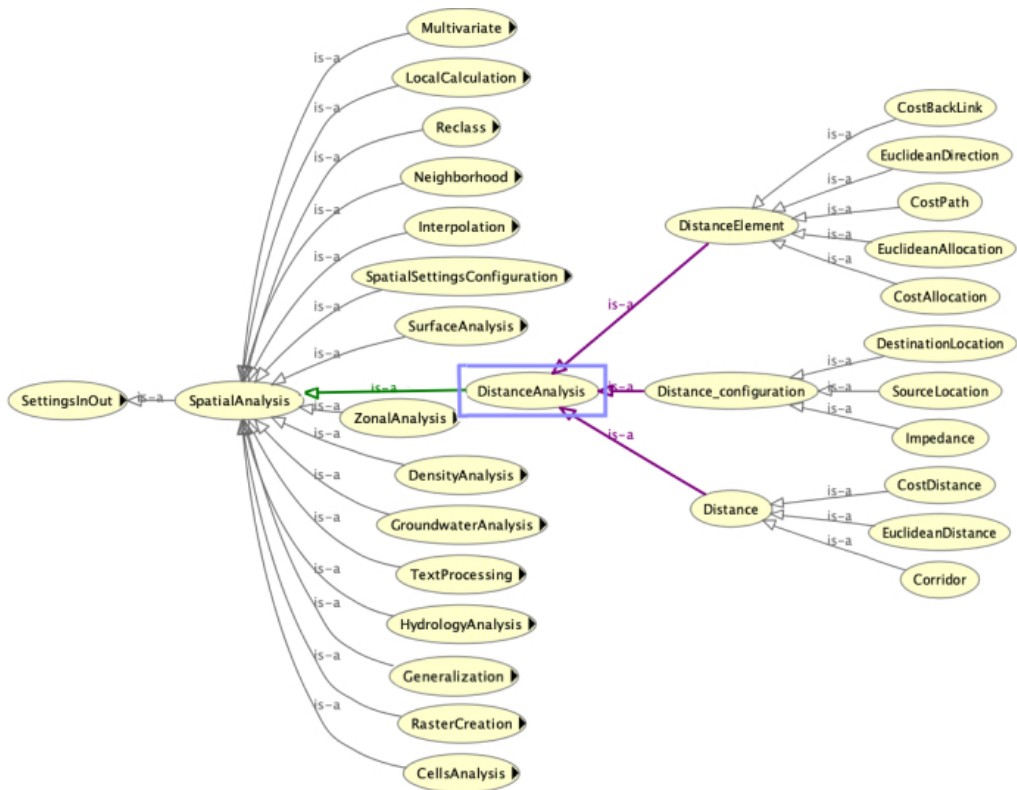

**Figure 8.** Extract from the data ontology for the *R1* request example.

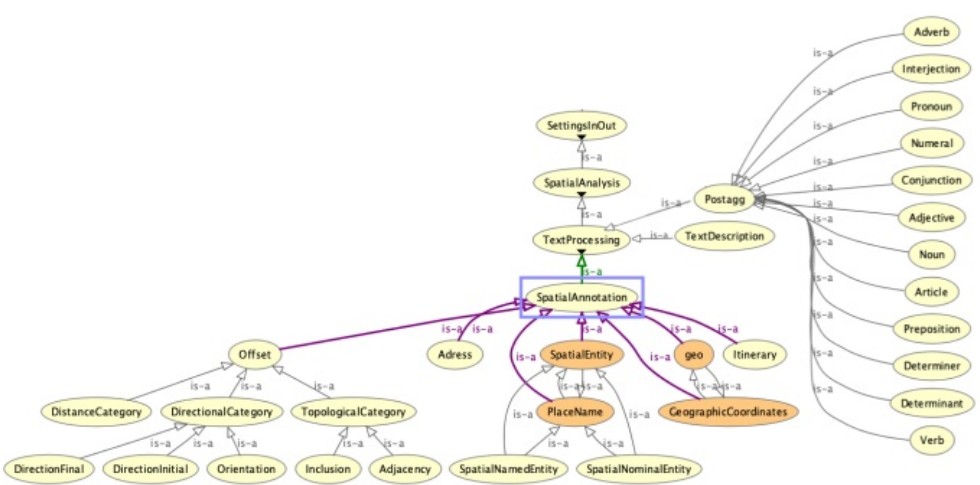

**Figure 9.** Extract from the data ontology for the *R2* request example.

4.3.2. Observations on the Experimentation Results

In the Configuration with Two Steps

The evaluation based on the set of tests shows that the discovery approach offers the highest precision rates (90%), recall rates (100%) and response time (10 s) for a functionality and I/O threshold of 0.5.

The highest recall rate (100%) is achieved for the threshold configurations {functionality threshold = 0.5, I/O threshold = 0.5} and {functionality threshold = 0.0, I/O threshold = 0.5}.

In contrast, in the case of the functionality threshold = 0.0, which simulates the non-consideration of the functionality matching step, the precision rate is not optimum (75%) and the response time increases significantly (83 s). This shows the importance of our first semantic matching step, which quickly excludes a large number of irrelevant services,

as shown in the results of request example *R2* (see Table 6) concerning the two services `ConvPerdido` and `Autocompletion`. Thus, this step reduces false positives by 15% and decreases the response time by 73 s. The highest precision rate (90%) is obtained for the threshold configurations {functionality threshold = 0.5, I/O threshold = 0.5} and {functionality threshold = 1.0, I/O threshold = 0.5}. However, in the case of functionality threshold = 1.0, which simulates a strict rigid semantic matching of the functionality, the recall rate is decreased (79%). In addition, if we apply a strict matching of I/O (I/O threshold = 1.0), this leads to a decrease in the recall rate, as shown in the results of request example *R1* (see Table 5). This provides an overview of the important role played by the proposed hybrid semantic matching method that allows us to compute numerical matching scores for the functionality and the I/O while avoiding drawbacks such as the principle of promoting neighboring concepts or the impossibility of distinguishing between the type of parameters to be matched. However, the precision rate includes a low percentage of false positives (10%). According to our analysis of the different requests used in the set of tests, false positives may occur in the case where all of the inputs required by the service are provided but a large number of outputs required by the request are not provided by the service. The reason for this is due to giving a similar weighting to the semantic matching score of inputs and outputs (see Equation (3a)). In the future, for a better satisfaction of the user's needs, it should be possible for the user to set the weighting values in order to calculate the global semantic correspondence score of I/O.

In the Configuration with Three Steps

We have previously mentioned the advantage of the proposed approach to compute a semantic matching score that allows us to rank the discovered services in a relevant order. However, in some cases, the scores (functional scores calculated on the I/O parameters) after the two primary steps are similar, resulting in sets of equivalent services without a preferred order. We can observe this on the tables presenting the results of the *R1* (see Table 5) and *R2* (see Table 6) requests. Indeed, considering a threshold = 0.5 for the functionality matching and the I/O matching, we see two sets of equivalent services for R1 (score of 0.97 and score of 0.72) and one set for *R2* (score of 1.0). In this context, the third matching steps try to refine the results and classify the services by calculating a global score, called "recommendation score", according to both functional and non-functional scores (see Equation (5)). To calculate the non-functional scores, two weightings $\alpha$ and $\beta$ are considered (see Equations (4) and (5)). In this experiment, $\alpha$ was set to 1.0 in order to consider only objective properties and $\beta$ to 0.5 for an equal consideration of functional and non-functional scores. Only objective properties were considered due to the low representation of subjective properties in the service corpus.

The objective properties include generic properties and specific geospatial services properties; their values are expressed in the numerical range [0, 1], unlike the subjective properties, where we opted for a star system (from one to five stars), to which, a normalization is then applied in order to obtain a score between [0, 1]. For example, for the objective geospatial property "currency", which represents the degree to which the source has incorporated changes (the frequency of updating), it can have five possible values—never, annual, monthly, weekly and daily—that will correspond to the numerical values 0, 0.25, 0.5, 0.75 and 1. The values of the properties are most often simulated in a more or less random way. This is the case for the services presented in response to the *R1* request. However, for the services related to the *R2* request, we assume that the values of generic objective properties are similar for the five selected services and we compute the values of objective properties specific to the geocoding functionality (see Table 1). Tables 7 and 8 show the recommendation scores for request *R1* and request *R2*, respectively (as mentioned in Section 3.2.2, remember that the functional score will be represented by the last computed score, which is the I/O score.).

**Table 7.** Recommendation score associated to the request *R1*.

| Service | Functional Score | Non-Functional Score | Recommendation Score |
|---|---|---|---|
| Euclidean Distance | 0.97 | 0.95 | 0.96 |
| Euclidean Allocation | 0.97 | 0.95 | 0.96 |
| Euclidean Direction | 0.97 | 0.90 | 0.93 |
| Cost Distance | 0.72 | 0.95 | 0.83 |
| Cost Allocation | 0.72 | 0.95 | 0.83 |
| Cost BackLink | 0.72 | 0.73 | 0.73 |

**Table 8.** Recommendation score associated to the request *R2*.

| Service | Functional Score | Non-Functional Score | Recommendation Score |
|---|---|---|---|
| GeoNames | 1.0 | 0.90 | 0.95 |
| GetToponyms * | 1.0 | 0.90 | 0.95 |
| TGN | 1.0 | 0.82 | 0.91 |
| SwissNames | 1.0 | 0.77 | 0.88 |
| OS50k | 1.0 | 0.72 | 0.86 |

* We consider the variant of the GetToponyms service that uses the GeoNames gazetteer.

In both cases, the recommendation scores refined the equivalent services sets. For the *R1* results, each service set was divided into two subsets, whereas, for *R2*, the initial set was divided into four subsets, which allows for a better distinction of services. It can be assumed that the result is better for *R2* because the specific properties are very discriminating, since the generic properties have the same values for all services. In addition, the values of the specific properties are real values, which makes the result more meaningful. These examples show that taking into account non-functional properties, even in a secondary way, brings a real added value to the recommendation of GWS.

**5. Conclusions**

The main research question behind this paper is the improvement of GWS discovery. This is of importance in the context of the Choucas project, where it is assumed that an improved access and processing of geospatial data, including a cross-analysis of heterogeneous data from multiple sources, should improve the efficiency of the mountain rescue process. GWS discovery is the process of selecting one or more services that meet a user's needs, taking into account functional and non-functional properties. A search is performed in a directory or catalogue of services in response to a user request. Using similarity measures, matching operations are performed between the parameters of the request and the properties stored in the service descriptions. Two main issues have been identified in GWS discovery processes. The first is the lack of semantics in the description of GWS, especially in the current standards. The second is the lack of consideration of advances in semantic matching, particularly in similarity calculation methods.

These issues are closely related because, if no semantic description is provided, the matching process is unable to exploit semantic knowledge, and semantic descriptions are meaningless without a semantic matching process to exploit them. Therefore, the contribution presented in this paper is twofold: (a) a description method based on a lightweight meta-model and a semantic annotation approach, and (b) a matching method applying a new semantic similarity measure on the service descriptions resulting from the application of the above mentioned method. The meta-model is derived from an abstraction of the GWS semantic description and has been instantiated in the OGC WPS standard. The interest is to provide valuable semantic information using metadata tags, while remaining compliant with the WPS standard. Our matching method is based on a three-step process. The first

step is the semantic matching of service functionality, the second is the semantic matching of I/O parameters and the third is the matching of non-functional properties. The core of this approach lies in the definition of *SimCalc*, a new hybrid matchmaking method for measuring the semantic similarity of WPS. An experiment was conducted on a corpus of 94 services with different characteristics: functionalities, types of I/O parameters, number of I/O parameters, etc. The results of this experiment are promising. In particular, they show the value of the three-step matching process. The first step avoids returning a large number of irrelevant services to the user. The second step refines the results on the selected services. Both steps involve the proposed *SimCalc* method to match the functional properties of the services, functionality for step 1 and I/O for step 2, with the parameters of the request. The non-logical part of the method calculates numerical matching scores between properties and parameters, whereas the logical part exploits individual links between properties and parameters to reduce false positives that may result from non-logical processing. To perform our experiments, we varied minimal acceptance thresholds. The evaluation based on the test set shows that the discovery approach offers the highest precision rates (90%), recall rates (100%) and response times (10 s) for a functionality and I/O threshold of 0.5. This can be considered an interesting improvement over approaches that do not consider the functionality matching step (where the precision rate is not optimal (75%) and the response time increased significantly (83 s)) and approaches that use strict semantic matching (where the recall rate is decreased to (79%)), which were simulated by varying the acceptance thresholds. These experiments have also shown that, in some cases, the functional semantic matching scores can be identical and do not allow us to classify the services. We then proceeded to a third step for the calculation of recommendation scores with a consideration of functional and non-functional scores. The examples presented above in Section 4.3.2 show that the recommendation scores have helped to refine the ranking of the services rendered, which allows for a better distinction of the services.

The presented work tends to improve the discovery of GWS by integrating semantic considerations at the description and semantic matching levels. On the description side, it allows for a WPS-compliant description to integrate the semantic description of functional and non-functional properties of services. On the semantic matching side, the *SimCalc* similarity calculation method takes into account advances in semantic matching by combining the advantages of the *SimWP* measurement, which is based on concept depth in a hierarchical structure, and the advantages of individual logical matching, which is based on fine-grained reasoning. Further experiments need to be conducted to confirm the first results. The third step has to be developed and evaluated on the basis of real values for non-functional properties. Therefore, in future work, we plan to develop a more advanced service recommendation process. On the one hand, by enriching the meta-model with non-functional properties (specific properties of particular geospatial services) that can be described and then annotated with an ontology to unify this knowledge in the same philosophy as for functional properties. Subsequently, regarding the calculation of the non-functional score related to objective properties, we also plan to integrate a finer order of preference between the different properties (as recommended by [39]). On the other hand, by taking into account the user profile and a previous selection context (as recommended by [31]).

**Author Contributions:** The authors state that the work was carried out in collaboration with all members. Meriem Sabrine Halilali: methodology, experimental work and writing; Eric Gouardères: methodology, supervision and writing; Mauro Gaio: methodology, supervision and writing; Florent Devin: review and involved in the discussion. All authors have read and agreed to the published version of the manuscript.

**Funding:** This document is the results of the research project Choucas funded by the French National Research Agency (ANR).

**Institutional Review Board Statement:** Not applicable.

**Informed Consent Statement:** Not applicable.

**Data Availability Statement:** The first version of the code supporting this publication is publicly available under the MIT license at https://github.com/ANRChoucas/Semantic-GWS-discovery. In order to test the discovery method in a more user-friendly way, we proposed a Java GUI available in the shared GitHub link.

**Conflicts of Interest:** The authors declare no conflict of interest.

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
