# Peer review of "Geospatial Web Services Discovery through Semantic Annotation of WPS"

_ijgi, doi:10.3390/ijgi11040254_

Round 1

Reviewer 1 Report

The contribution of this paper is about improving discovery of Geospatial Web Services (GWS) by integrating semantic aspects in the model of service and exploiting them for semantic matching with service requests. The proposed semantic characterization is based on annotating service functionalities with concepts from a functionality ontology and service I/O parameters with a data ontology. The functional-based modeling and discovery is integrated with non-functional characterization of services.

On the positive side:

- the paper is generally clear and quite well written

- experimentation based on evaluating classical precision and recall metrics over a dataset of 94 services is provided

- a GitHub repository with the source code of the prototype, the semantic descriptions of services of the used dataset, and the ontologies, is provided. A dataset of annotated GWS is made available and this is something really valuable.

On the negative side:

- the main ideas of the approach are well known in the literature. Semantic annotation of web services, semantic augmentations for Geospatial Services, service matchmaking based on the service functionality and I/Os have been widely discussed, also including descriptions ad use of non-functional features.

E.g.,

Cheng Y., Ge W., Xu L. (2018) Quality of Geographical Information Services Evaluation Based on Order-Relation. In: Zhou Q., Gan Y., Jing W., Song X., Wang Y., Lu Z. (eds) Data Science. ICPCSEE 2018. Communications in Computer and Information Science, vol 901. Springer, Singapore. https://doi.org/10.1007/978-981-13-2203-7_56

Yue, P., Di, L., Zhao, P., Yang, W., Yu, G., & Wei, Y. (2006, July). Semantic augmentations for geospatial catalogue service. In 2006 IEEE International Symposium on Geoscience and Remote Sensing (pp. 3486-3489). IEEE.

Bianchini, D., De Antonellis, V. & Melchiori, M. Flexible Semantic-Based Service Matchmaking and Discovery. World Wide Web 11, 227–251 (2008). https://doi.org/10.1007/s11280-007-0040-y

Therefore, the paper should be motivated and characterized better in order to be accepted as a journal publication.

- the presented approach seems completely manual. That is, it requires a lot of effort for real large service catalogues.  In the recent research, there are works trying to semi-automate this task (e.g., Saquicela, V., Vilches-Blázquez, L. M., Freire, R., & Corcho, O. (2022). Annotating OGC web feature services automatically for generating geospatial knowledge graphs. Transactions in GIS, 26, 505– 541. https://doi.org/10.1111/tgis.12863) but the presented approach does not address this relevant point.

- how the proposed ontologies (service ontology and data ontology) have been designed? A relavant point in designing an ontology is the amount of reuse of (parts of) existing ontologies that is done and how it is linked to them. This point seems it is not addressed in the paper.

- integration of non-functional features in the matchmaking model is not clear. How service subjective and objective properties are mapped to a number (the value of the objective property) in the range [0..1]?

- the usage of the GUI of the prototype in order to perform annotation and discovery should be illustrated in the paper.

Author Response

Dear Reviewer,

First of all, on behalf of all authors, I would like to thank you for spending time to review our work. Please find attached the detailed responses to your comments/questions.

Respectfully yours,

Meriem Sabrine HALILALI

Reviewer 2 Report

Dear colleagues, thank you for your important work on "geospatial web services discovery through semantic annotation of WPS". In times when spatial communities calls Service-Oriented Architectures in question, because of its implementation efforts - especially the detailed creation of metadata - this contribution is able to show the perspectives of the service oriented approach as soon as a Geospatial Web Service corpus exists and the requirements for the functional-, non-functional matching are sufficiently considered. 

I am just having some questions:

In line 72 you highlight taht ISO 19119:2016 and OGC WPS 2.0 do not yet include the possibility of providing semantic information in the service description. Are you involved in working groups that are able to extend existing standards? Are there any intentions for extending the standards? Are the requirements, which are described in your paper, communicated into the standard adopting working groups?

In line 289 you mention that the described method accepts only exact and subsumes filters to match these parameters. Could this be seen as disadvantage or advantage of this specific method? Is there a way to make a better use of this characteristic?

Thank you and I am looking forward to the next results of your work...

Author Response

(The authors gave the same response as above.)

Reviewer 3 Report

Page 9 : In Eq (2), different from the 4th, 5th, and additional cases, the 2nd and 3rd cases have their own function(SimWP(SP, RP)), respectively. It seems better the 4th, 5th, and additional cases have the resulting values (0.0), respectively. 

Page 10 : if there are references for Eq(3) and Eq(4), would please present them.

Author Response

(The authors gave the same response as above.)

Round 2

Reviewer 1 Report

I thanks the Authors for the good work in improving the paper and for the convincing answers to my questions. 

I would just recommend again to the Authors to consider the opportunity of including at least a screenshot of the main window of GUI of the tool they developed.